# Does 'Deep Learning on a Data Diet' reproduce? Overall yes, but GraNd at Initialization does not

**Andreas Kirsch**                                                                *andreas.kirsch@cs.ox.ac.uk*
*OATML, Department of Computer Science*
*University of Oxford*

**Reviewed on OpenReview:** *https://openreview.net/forum?id=1dwXa9vmOI*

## Abstract

Training deep neural networks on vast datasets often results in substantial computational demands, underscoring the need for efficient data pruning. In this context, we critically re-evaluate the data pruning metrics introduced in 'Deep Learning on a Data Diet' by Paul et al. (2021): the *Gradient Norm (GraNd)* (at initialization) and the *Error L2 Norm (EL2N)*. Our analysis uncovers a strong correlation between the GraNd scores at initialization and a sample's input norm, suggesting the latter as a potential baseline for data pruning. However, comprehensive tests on CIFAR-10 show neither metric outperforming random pruning, contradicting one of the findings in Paul et al. (2021). We pinpoint the inconsistency in the GraNd at initialization results to a later-fixed bug in FLAX's checkpoint restoring mechanism[1]. Altogether, our findings do not support using the input norm or GraNd scores at initialization for effective data pruning. Nevertheless, EL2N and GraNd scores at later training epochs do provide useful pruning signals, aligning with the expected performance.

## 1 Introduction

Deep neural networks have achieved state-of-the-art performance on tasks like image classification and text generation, but require massive datasets that demand extensive computational resources for training (Deng et al., 2009; Brown et al., 2020). This motivates developing techniques like data pruning to identify effective subsets of the training data (Bachem et al., 2017; Sorscher et al., 2022).

Recently, 'Deep Learning on a Data Diet' (Paul et al., 2021) introduced two new pruning metrics, *Gradient Norm (GraNd)* at initialization and *Error L2 Norm (EL2N)*. During a talk that was the given on the paper, the surprising effectiveness of GraNd at initialization—before any training—led us to hypothesize on a potential correlation with a sample's input norm. The input norm is cheap to compute and could thus provide an intriguing new baseline for data pruning experiments.

In this work, we set out to reproduce the results for GraNd scores at initialization and explore the practicality of using the input norm as a proxy. Through extensive analysis on CIFAR-10, we find a strong correlation between the input norm and the gradient norm at initialization. However, neither metric outperform random pruning, failing to reproduce the original findings. While we replicate the reported results for EL2N and GraNd scores at later epochs, the performance of GraNd scores at initialization is not reproducible despite testing with multiple codebases. We trace this back to a bug in the checkpoint restoring code of the underlying FLAX library. By rectifying the claims about GraNd scores at initialization and uncovering limitations of the input norm, our reproduction provides updated recommendations for efficient data pruning.

In summary, this reproduction contributes a new insight on the relationship between the input norm and the gradient norm at initialization and fully explains a failure to reproduce the performance of using the gradient norm at initialization for pruning, one of the six contributions of Paul et al. (2021).

---

[1]Details at `https://github.com/google/flax/commit/28fbd95500f4bf2f9924d2560062fa50e919b1a5`.

**Outline.** In §2, we provide background on the GraNd and EL2N scores and discuss their use for data pruning. In §3.1, we begin by discussing the correlation between input norm and gradient norm at initialization. We empirically find strong correlation between GraNd scores at initialization and input norms as we average over models. In §3.2, we explore the implication of this insight for dataset pruning and find that both GraNd at initialization and input norm scores do not outperform random pruning, but GraNd scores after a few epochs perform similar to EL2N scores at these later epochs.

## 2 Background

**'Deep Learning on a Data Diet'.** Paul et al. (2021) introduce two novel metrics: *Error L2 Norm (EL2N)* and *Gradient Norm (GraNd)*. These metrics aim to provide a more effective means of dataset pruning. It is important to emphasize that the GraNd score at initialization is calculated before any training has taken place, averaging over several randomly initialized models. This fact has been met with skepticism by reviewers[2], but Paul et al. (2021) specifically remark on GraNd at initialization:

> **Pruning at initialization.** In all settings, GraNd scores can be used to select a training subset at initialization that achieves test accuracy significantly better than random, and in some cases, competitive with training on all the data. This is remarkable because GraNd only contains information about the gradient norm at initializion, averaged over initializations. This suggests that the geometry of the training distribution induced by a random network contains a surprising amount of information about the structure of the classification problem.

**GraNd.** The GraNd score measures the magnitude of the gradient vector for a specific input sample in the context of neural network training over different parameter draws. The formula for calculating the (expected) gradient norm is:

$$\text{GraNd}(x) = \mathbb{E}_{\theta_t}[\|\nabla_{\theta_t} L(f(x; \theta_t), y)\|_2] \tag{1}$$

where $\nabla_{\theta_t} L(f(x; \theta_t), y)$ is the gradient of the loss function $L$ with respect to the model's parameters $\theta_t$ at epoch $t$, $f(x; \theta)$ is the model's prediction for input $x$, and $y$ is the true label for the input. We take an expectation over several training runs. The gradient norm provides information about the model's sensitivity to a particular input and helps in identifying data points that have a strong influence on the learning process.

**EL2N.** The EL2N score measures the squared difference between the predicted and (one-hot) true labels for a specific input sample, also known as the Brier score (Brier, 1950):

$$\text{EL2N}(x) = \mathbb{E}_{\theta_t}[\|f(x; \theta_t) - y\|_2^2], \tag{2}$$

where $f(x; \theta)$ is the model's prediction for input $x$, $y$ is the (one-hot) true label for the input, and $\|\cdot\|_2$ denotes the Euclidean (L2) norm. The EL2N score provides insight into the model's performance on individual data points, allowing for a more targeted analysis of errors and potential improvements.

The GraNd and EL2N scores are proposed in the context of dataset pruning, where the goal is to remove less informative samples from the training data. Thus, one can create a smaller, more efficient dataset that maintains the model's overall performance while reducing training time and computational resources.

While GraNd at initialization does not require model training, it requires a model and is not cheap to compute. In contrast, the input norm of training samples is incredibly cheap to compute and would thus provide an exciting new baseline to use for data pruning experiments. We investigate this correlation next and find positive evidence for it. However, we also find that the GraNd score at initialization does not outperform random pruning, unlike the respective results in Paul et al. (2021).

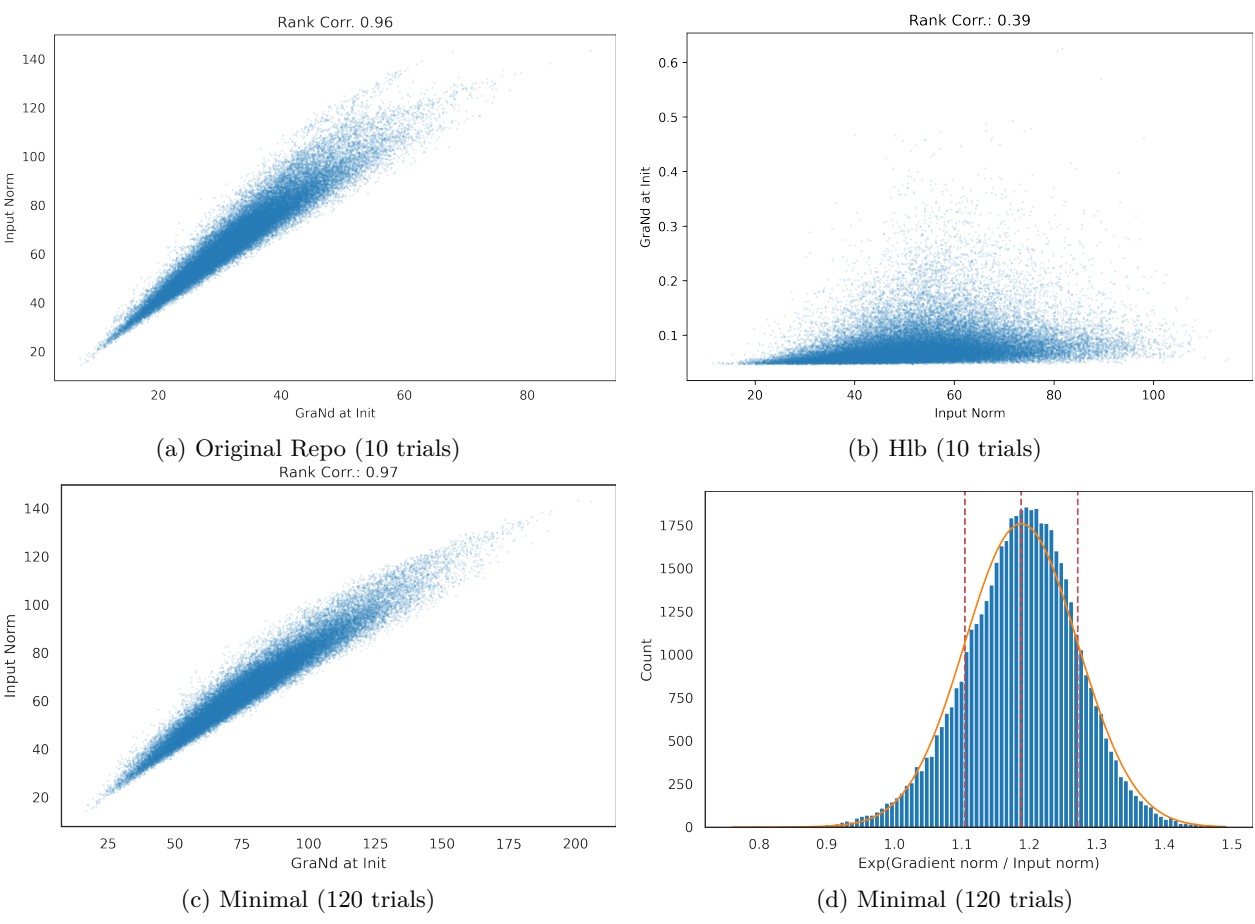

Figure 1: *Correlation between GraNd at Initialization and Input Norm for CIFAR-10's training set.* (**a**, **b**, **c**): The original repository and the 'minimal' implementation have very high rank correlation—'hlb' has a lower but still strong rank correlation. (**d**): *Ratio between input norm and gradient norm.* In the 'minimal' implementation, the ratio between input norm and gradient norm is roughly log-normal distributed.

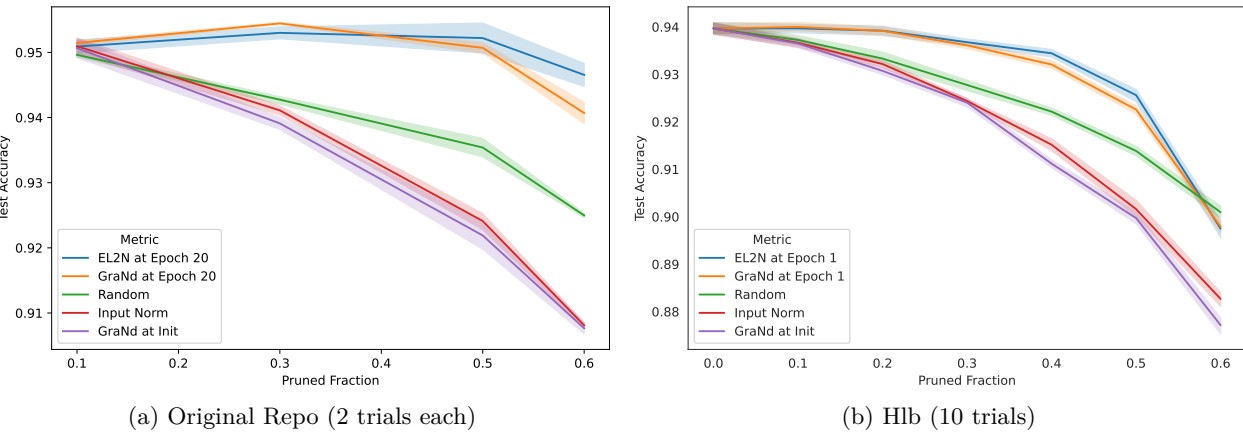

Figure 2: *Reproduction of Figure 1 (second row) from Paul et al. (2021).* In both reproductions, GraNd at initialization performs as well as the input norm. However, it does not perform better than random pruning. Importantly, it also fails to reproduce the results from Paul et al. (2021). However, GraNd at epoch 20 (respectively at epoch 1 for 'hlb') performs similar to EL2N and like GraNd at initialization in Paul et al. (2021).

## 3 Investigation

We examine the correlation between the input norm and GraNd at initialization, as well as other scores on CIFAR-10 (Krizhevsky et al., 2009), through three distinct approaches:

1. First, we update the original paper repository[3] (https://github.com/mansheej/data_diet), which utilizes JAX (Bradbury et al., 2018). We rerun the experiments for Figure 1 (second row) in Paul et al. (2021) on CIFAR-10, training for 200 epochs using GraNd at initialization, GraNd at epoch 20, E2LN at epoch 20, Forget Score at epoch 200, and input norm.

2. Second, we reproduce the same experiments using 'hlb' (Balsam, 2023), a significantly modified version of ResNet-18 that achieves high accuracy in 12 epochs, taking approximately 30 seconds in total on an Nvidia RTX 4090 with PyTorch (Paszke et al., 2019). For this approach, we compare GraNd at initialization, GraNd at epoch 1 ($\approx 20/200 \cdot 12$ epochs), EL2N at epoch 1, and input norm[4]. We analyze the rank correlations between the different scores for the two repositories mentioned above.

3. Third, we employ another 'minimal' CIFAR-10 implementation (van Amersfoort, 2021) with a standard ResNet18 architecture for CIFAR-10 to compare the rank correlations.

### 3.1 Correlation between GraNd at Initialization and Input Norm

To gain a deeper understanding of the relationship between the input norm and the gradient norm at initialization, we first consider a toy example and then provide empirical evidence. Let us examine a linear softmax classifier with $C$ classes (without a bias term). The model takes the form:

$$f(x) = \text{softmax}(Wx), \tag{3}$$

together with the cross-entropy loss function:

$$L = -\log f(x)_y. \tag{4}$$

The gradient of the loss function concerning the rows $w_j$ of the weight matrix $W$ is:

$$\nabla_{w_j} L = (f(x)_j - \mathbb{1}\{j = y\})x, \tag{5}$$

where $\mathbb{1}\{j = y\}$ is the indicator function that is 1 if $j = y$ and 0 otherwise. The squared norm of the gradient is:

$$\|\nabla_w L\|_2^2 = \sum_{j=1}^{C} (f(x)_j - \mathbb{1}\{j = y\})^2 \|x\|_2^2. \tag{6}$$

Taking expectation over $W$ (different initializations), the norm of the gradient is:

$$\mathbb{E}_W \left[ \|\nabla_w L\|_2 \right] = \mathbb{E}_W \left[ \left( \sum_{j=1}^{C} (f(x)_j - \mathbb{1}\{j = y\})^2 \right)^{1/2} \right] \|x\|_2. \tag{7}$$

As a result, the gradient norm is a multiple of the input norm. The factor depends on $f(x)_j$, which we could typically expect to be $1/C$ in expectation over different weights at initialization (due to class symmetry).

---

[2]See also https://openreview.net/forum?id=Uj7pF-D-YvT&noteId=qwy3HouKSX.
[3]See https://github.com/blackhc/data_diet
[4]https://github.com/blackhc/pytorch_datadiet

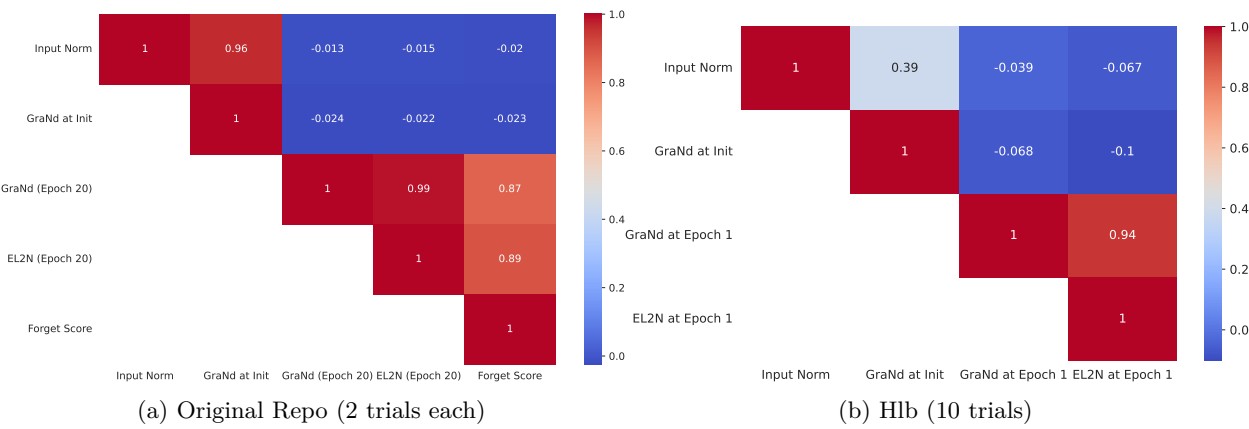

(a) Original Repo (2 trials each)       (b) Hlb (10 trials)

Figure 3: *Rank Correlations of the Scores.* Cf. Figure 12 in the appendix of Paul et al. (2021). In both reproductions, GraNd at initialization and input norm are positively correlated, while GraNd and EL2N at later epochs are strongly correlated with each other and the Forget Score (at epoch 200).

## 3.2 Empirical Results

**Correlation Analysis.** We first examine the correlation between the input norm and GraNd score at initialization. Across three separate codebases and implementations (original JAX, 'hlb' PyTorch, 'minimal' PyTorch), we consistently find a strong Spearman rank correlation between the two metrics when averaged over multiple random initializations (Figure 1). For example, the original JAX and 'minimal' PyTorch implementation have a rank correlation coefficient of 0.96 between input norm and GraNd on the CIFAR-10 training set over 10 trials, suggesting input norm could potentially serve as an inexpensive proxy for GraNd at initialization. 'hlb' uses different input preprocessing and other tricks, which might explain the lower but still strong rank correlation coefficient of 0.39.

**Reproducing Figure 1 of Paul et al. (2021) on CIFAR-10.** However, our attempts to reproduce the data pruning results originally reported for GraNd at initialization are unsuccessful. As shown in Figure 2, neither input norm nor GraNd at initialization outperform random pruning, with both achieving approximately 2 percentage points less accuracy than random pruning when pruning 60% of the CIFAR-10 training data. This directly contradicts the original findings of Paul et al. (2021), where GraNd at initialization markedly exceed random pruning when pruning 60% of the CIFAR-10 training data. While we are able to reproduce the expected pruning performance for GraNd after 20 epochs of training (respectively at epoch 1 for 'hlb') and also match the original EL2N results, GraNd at initialization does not match expectations despite testing across codebases.

**Score Rank Correlations.** To further analyze relationships between the different scores, we visualize their Spearman rank correlations on CIFAR-10 using a correlation matrix (Figure 3). The results confirm the strong correlation between input norm and GraNd at initialization. They also reveal high correlations between GraNd at later epochs, EL2N, and the Forget Score after full training. However, GraNd at initialization and input norm show little correlation with the later epoch scores. This aligns with our reproduction results, where only the late-epoch GraNd and EL2N scores exhibit the expected pruning performance. The distinct correlation profile provides additional evidence that GraNd at initialization is measuring something fundamentally different from the trained model pruning metrics.

## 4 Discussion

While input norm is inexpensive to compute, given its model independence and lower computational cost compared to GraNd or other scores, our results imply it should not be used for data pruning. Similarly, since only GraNd scors at later epochs appears to perform as expected, we cannot recommend using GraNd scores at initialization for data pruning either.

Regarding the failure to reproduce the GraNd at initialization results, we initially could not rerun the code using the original JAX version due to GPU incompatibility. However, the authors of Paul et al. (2021) managed to reproduce their original results by setting up an old VM image with the original dependencies. This led us to discover a later-fixed bug in the FLAX checkpoint restore function `flax.training.restore_checkpoint`[5] as the underlying cause of the reported GraNd at initialization results: the code was restoring different checkpoints than expected. Our experience reinforces the importance of preserving code environments and dependencies to enable reproducibility.

Our findings highlight the critical need for reproductions, especially when initially surprising results are reported. As state-of-the-art techniques become more complex, it is essential that different research groups thoroughly verify results across frameworks and implementations before informing best practices.

## 5 Conclusion

This work makes multiple contributions. We uncovered a strong correlation between the inexpensive input norm and the proposed GraNd metric at initialization. However, through extensive analysis we found neither input norm nor GraNd scores at initialization reproduced the originally reported pruning performance. Only GraNd and EL2N scores at later training epochs provided useful pruning signals, aligning with expected behavior. We traced the non-reproducible GraNd at initialization results to a now-fixed bug in the checkpoint restoring code of the underlying FLAX framework. Our findings rectify claims around GraNd scores at initialization, provide updated recommendations on utilizing these data pruning metrics, and underscore the importance of rigorously confirming initial surprising results across implementations. As deep learning advances, maintaining high standards for reproducibility will ensure progress builds on solid foundations.

### Acknowledgments

Many thanks to Mansheej Paul and Karolina Dziugaite for very helpful feedback, discussions, and in particular, for their assistance in investigating the source of the discrepancy in reproducing the GraNd at initialization results. Upon being informed of our findings, they worked to trace the issue and shared their methodology for reproducing the original results using the original framework versions. This helped uncover the checkpoint restoring bug as the underlying cause. We appreciate that they took the time to rerun experiments with the original codebase and dependencies, and for quickly preparing an updated version of their paper.

AK is supported by the UK EPSRC CDT in Autonomous Intelligent Machines and Systems (grant reference EP/L015897/1). ChatGPT (GPT-4) and Claude 2 were used to suggest edits and improvements.

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

# A   Appendix

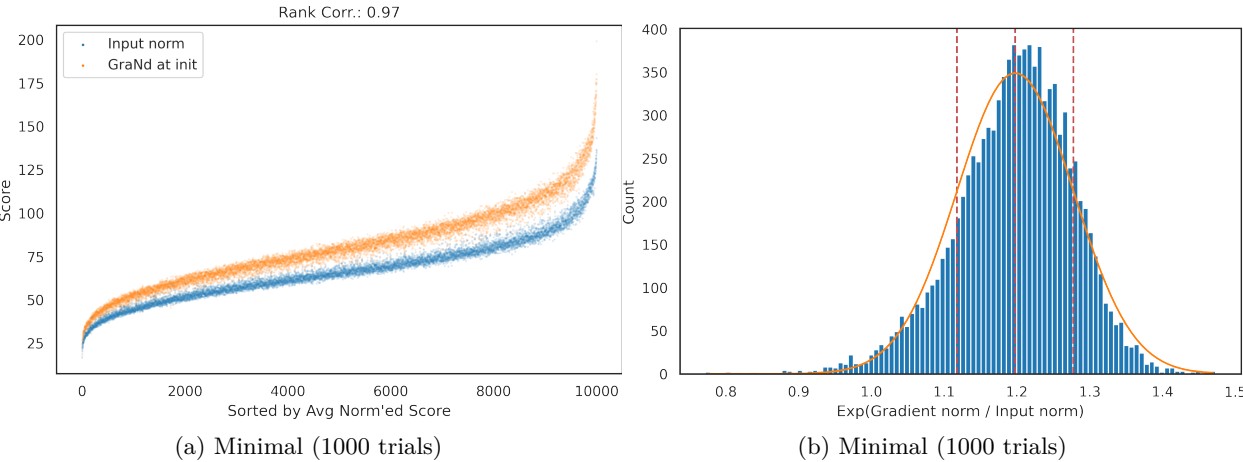

(a) Minimal (1000 trials)  (b) Minimal (1000 trials)

Figure 4: *Correlation between GraNd at Initialization and Input Norm on the Test Set.* (**a**): We sort the samples by their average normalized score (i.e., the score minus its minimum divided by its range), plot the scores and compute Spearman's rank correlation on CIFAR-10's test data. The original repository and the 'minimal' implementation have very high rank correlation—'hlb' has a lower but still strong rank correlation. (**b**): *Ratio between input norm and gradient norm.* In the 'minimal' implementation, the ratio between input norm and gradient norm is roughly log-normal distributed

