# OpenReview forum: "Does ‘Deep Learning on a Data Diet’ reproduce? Overall yes, but GraNd at Initialization does not"
_TMLR — Accepted by TMLR_

### Review · Reviewer_LnNb · 2023-05-01

**Summary Of Contributions:**

This work investigates whether one of the key metrics (GraNd) introduced by the Data-Diet paper is indeed useful in dataset pruning, as reported in that paper. Through thorough experiments on two architectures using both the original repo and a new reimplementation, the authors find that GraNd cannot outperform random pruning, while the other metric (EL2N) works as reported. The authors further reported that a bug fix in the supporting training framework of Data-Diet repo (JAX) is likely the reason for the inconsistency.

**Audience:**

Yes

**Broader Impact Concerns:**

no clear negative society impacts from this.

**Claims And Evidence:**

Yes

**Requested Changes:**

See the weakness.

**Strengths And Weaknesses:**

Strengths:
The authors provided a clear motivation for why they started checking whether GraNd (norm of the gradients at the initialization step) is useful, which is due to its strong correlation to the input-norm metric and therefore may lead to a dataset pruning metric that is both cheap to compute and powerful.
The authors also used both the original repo and a re-implementation repo to confirm that the failure in reproducing the result is not due to an error in their implementation.
The strong correlation between the input-norm and the GraNd metrics also indicates that this GraNd metric is unlikely to work as reported in the DataDiet paper, as it would be difficult to imagine that the input norm would be important in deciding how good the example is for training.
Finally, the authors also identified the possible cause of the original successful result in the Data-Diet paper.
Taken together, these results help the community further explore which metric is important for data pruning.

Weaknesses:
The first figure is a bit confusing. If the authors want to show the strong correlation between the two metrics (input-norm and GraNd), why not show a scatter plot with X-axis being one metric and Y-axis being another metric?
The writing can be improved. It’s currently informal and can be clearer. For example, the abstract is a bit influent in its logic, with the motivation not clearly explained before the result.

---

> ### Author Response · Authors · 2023-05-29
> **Thank you!**
>
> Dear Reviewer LnNb,
>
> Thank you for your thoughtful review of our paper and for providing constructive feedback. We appreciate your positive remarks on the clear motivation and the thorough investigation of the GraNd metric in our work.
>
> In response to your suggestions, we have revised Figure 1, clarified the abstract, and edited the paper to improve the writing.
>
> Your comments have been very helpful. Please let us know if you are satisfied with our revision and any other feedback.
>
> Best regards,\
> the Authors

---

### Review · Reviewer_UzQW · 2023-05-08

**Summary Of Contributions:**

The paper reports on the failure to fully reproduce results of Paul et al, 2021. In particular, gradient at initialization (GranD) scoring did not perform better than random pruning of training examples. Authors used the original source code as well as another optimized ResNet-18 implementation. It seems that the cause for the non-reproducibility is a (now fixed) bug in the Flax library.

**Audience:**

Yes

**Broader Impact Concerns:**

No concerns

**Claims And Evidence:**

Yes

**Requested Changes:**

In my opinion, the second paragraph of the paper is not neutral to the principle of double-blind review and contains details that are irrelevant for the majority of readers. I request these details to be removed.

**Strengths And Weaknesses:**

*Strengths*
* The paper contributes to the community by reproducing and valdiating established results.

*Weaknesses*
* I don't know if, in this case, a TMLR paper is the best format for sharing the results. The issue seems to be a software bug, not a conceptual error. It's important to know that GraNd doesn't work at iteration 0, but it works at other iterations so unless Pauls et al _predicted_ that GraNd must have worked at iteration 0 (as I understand, that's not the case) correcting the original paper feels enough to me. I wonder if authors can suggest what could be the other consequences of their experimental findings.

---

> ### Author Response · Authors · 2023-05-29
> **Thank you!**
>
> Dear Reviewer UzQW,
>
> Thank you for your insightful comments and your recognition of our work's contribution to the community through validation and reproduction of established results.
>
> We appreciate your concerns about the paper's format. Although the issue ultimately stemmed from a bug in JAX/FLAX, now fixed, the implications for our understanding and application of the GraNd metric extend beyond a mere bug report.
> Identifying the bug was also not trivial and required considerable effort during our investigation. The initial findings regarding the effectiveness of GraNd at initialization had already prompted significant research on our end (to use the input norm as a new alternative baseline), further emphasizing the importance of informing others in the community of our findings and the necessity of their publication.
>
> In particular, the original paper version remarked on the following:
>
> > Pruning at initialization. In all settings, GraNd scores can be used to select a training subset at initialization that achieves test accuracy significantly better than random, and in some cases, competitive with training on all the data. *This is remarkable because GraNd only contains information about the gradient norm at initializion, averaged over initializations.* **This suggests that the geometry of the training distribution induced by a random network contains a surprising amount of information about the structure of the classification problem. EL2N scores, which only contain information about errors, are not consistently effective at initialization and forgetting scores, which require counting forgetting events over training, are not defined at initialization.**
>
> (Highlights added by us, see https://arxiv.org/pdf/2107.07075v1.pdf; see also other occurrences of "at initialization" in that paper revision.)
>
> Regarding the second paragraph in our work, we have made revisions, as you suggested, while preserving the essence of the anecdote. This anecdote provides essential context for our investigation: it emphasizes the surprising reported findings for GraNd at initialization in the original paper. These findings were reported widely and, as such, likely "updated" the assumed intuitions and understanding of many, which reinforces the relevance of our study now to correct the record.
>
> We note your concerns about the principle of double-blind review in relation to this anecdote. However, we do not believe that the anecdote discloses any information that would compromise the review process's anonymity. If you believe that specific elements within it could jeopardize this, we kindly ask you to point them out to us.
>
> Thank you once again for your valuable comments. We look forward to hearing whether our revision has adequately addressed your feedback.
>
> Best regards, \
> the Authors

---

### Review · Reviewer_iV8N · 2023-05-26

**Summary Of Contributions:**

This paper examines one of the novel approaches introduced by Paul et al. (2021) in their paper titled "Deep Learning on a Data Diet," known as the GradND. GradND is a method used to prune away unnecessary data in the training dataset for more effective training using the reduced dataset. In the present study, the authors shed light on a significant issue discovered within the experiments conducted by Paul et al. (2021), which has resulted in misleading experimental outcomes. Specifically, the authors of this paper show that the results of GradND obtained in the original paper by Paul et al. (2021) at initialization of the model is wrong and caused by a bug in their implementation.






**Audience:**

Yes

**Broader Impact Concerns:**

No concerns.

**Claims And Evidence:**

Yes

**Requested Changes:**

I don't think there are any changes needed.

**Strengths And Weaknesses:**

Strengths:
All in all, the paper is clearly written and the the statement seems to be technically sounds. I have noticed that Paul et al. (2021) has also updated their arxiv version of the "Deep Learning on a Data Diet," paper to account for the mistake pointed out.

Weaknesses:
As a paper solely submitted to point out a bug from a previous paper, I don't think there is much major weaknesses. The bug pointed out seems to be real and the conclusion from this paper seems to be correct. Nevertheless, I am not sure if such a paper is worth publishing, especially after Paul et al. (2021) has fixed the issues in their version of the paper.

---

> ### Author Response · Authors · 2023-05-29
> **Thank you!**
>
> Dear Reviewer iV8N,
>
> We appreciate your thorough review of our paper and your recognition of its technical soundness.
>
> Our investigation was inspired by the prospect and hope of using input norms as a surrogate for GraNd at initialization, which would have major implications given the original findings. While a JAX/FLAX bug was ultimately identified as the cause of these results, we argue that the wider consequences of this bug---a peer-reviewed publication reported specific findings in data pruning because of the bug, which they had to then correct---warrant a dedicated & published reproducibility study.
>
> Our work is of importance even in light of a subsequent correction of the original paper on arXiv, which has not undergone peer review, however, and has not been widely disseminated either---unlike the original paper accepted at a top-tier conference.
>
> Thank you for your feedback and for engaging with our work.
>
> Best regards, \
> the Authors

---

### Author Response · Authors · 2023-05-29
**On the publishability of reproducibility studies**

Dear Reviewers,

We sincerely thank you for your time and feedback. Given the concerns raised about our paper's publishability by two reviewers, we'd like to clarify our motivations, contributions, and the importance of our work within the broader context of machine learning research in the following unified response.

**Importance of Reproducibility Studies.** Reproducibility is a cornerstone of scientific research, ensuring the validity of findings, reinforcing the reliability of methods, and promoting transparency. This principle is particularly important in fast-paced fields like deep learning, where reproducibility studies can identify and rectify both empirical and conceptual errors, thereby strengthening the robustness of the algorithms and techniques.

TMLR specifically "*invites authors to submit papers that contain
[…] reproducibility studies of previously published results or claims;*" and also has a [Reproducibility](https://jmlr.org/tmlr/papers/#) award, which "*is awarded to papers whose primary purpose is reproduction of other published work. Beyond simple verification, the paper must contribute significant added value through additional baselines, analysis, ablations, or insights.*"

Our work goes beyond a simple reproduction using multiple code bases by identifying a non-obvious bug in the underlying deep learning framework of the original paper, which necessitated a correction and revision of the original paper and contributions.
Thus, it does more than merely point out a reproduction failure or question previous empirical results. We have established the precise causal reason for incorrect empirical results in a peer-reviewed paper published at a top-tier conference.
We thus provide definitive evidence that GraNd at initialization does not function as initially claimed.

This, we would argue, demonstrates significant added value.

**Specific Contribution of Our Paper.** Our paper investigates the GraNd at Initialization metric from the "Data Diet" paper. Our submission outlines the motivations behind this study:

1. We identified a potential correlation between GraNd at initialization and input norm, suggesting that the latter might have provided a simple yet efficient baseline, given that GraNd at initialization, which is more complex to compute, was reported to be highly effective.
2. The authors and reviewers of the original paper found the initial results surprising but plausible and henceforth reported them widely (in talks and on Twitter).
3. GraNd at initialization was an important contribution of the original paper, as also mentioned in our submission.

Given these points, our reproducibility study's results and contributions are highly significant, enhancing the community's understanding of a published dataset pruning metric. Readers of the original paper need to re-revise their knowledge and intuitions based on our findings.

**Response to the Notion that Corrected Papers Invalidate Reproducibility Studies.** A concern was raised regarding the relevance of our paper following the corrections made by the original authors.

These corrections, however, underscore the value and necessity of our work. Our findings prompting revisions in the original paper illustrate this reproducibility study's impact and relevance.

Furthermore, we reached out to the authors after observing reproducibility issues. This allowed them to revise their paper in a timely fashion. We believe that such cooperative approaches, while not obligatory, can foster healthier scientific discourse.

However, if corrections resulting from a study were to invalidate its later publication, it might lead to more adversarial interactions in the community. For one, we would need to reconsider future cooperative practices if they threatened our chances of publication, given the  effort we have invested in this project.

Overall, if reproducibility studies were deemed unpublishable due to subsequent or timely corrections by original authors, it would set a troubling precedent. This stance could deter future reproduction efforts, thereby undermining the rigour and trustworthiness of machine learning research.

Our paper, and others like it, are instrumental in maintaining a high standard of scientific integrity and should be acknowledged as such.
We trust that you recognize the value of our work within this broader context.

We have already incorporated your other feedback and are happy to revise our paper further, given any additional input from you. Again, thank you for your reviews and feedback and for engaging with our responses.

Best regards, \
the Authors

---

### Decision · Action_Editors · 2023-08-18

**Recommendation:** Accept as is

**Comment:**

The paper is written well. The experiments are convincing. The paper explains why it failed to replicate some of the experiments, and the results are interesting. Especially, I am quite surprised at how strong the random pruning performs. Also, the strong correlation between the GraND score and input norms makes sense. It is well explained too. The paper is short, but I think that is appropriate since the conclusions that can be drawn from it are straightforward and no need to be obfuscated.

**Audience:**

This paper would be of interest of TMLR's audience who might be interested in the data pruning studies and replication efforts.

**Claims And Evidence:**

This work analyses the 'Deep Learning on a Data Diet' paper and two different scores introduced in that paper (GraND and EL2N) for pruning datasets during the training of neural networks. The paper first tried to reproduce the results achieved in the original paper with those two scores. They successfully replicated the results for the EL2N score at epoch 20 but failed to replicate the experiments with the GraND score. After investigating this with the authors of the original paper, they realized that the results in the original paper were due to a bug related to checkpointing with flax.training.restore_checkpoint function in flax. The paper also showed a strong rank correlation between the GraND score at initialization and the input norm, which is much easier to compute. However, in the end, they showed that both GraND and EL2N could not outperform the simple baseline of random pruning.
The experiments are thorough, and the results are convincing.